# A Facile Method for the Generation of Fe_3_C Nanoparticle and Fe-N_x_ Active Site in Carbon Matrix to Achieve Good Oxygen Reduction Reaction Electrochemical Performances

**DOI:** 10.3390/ma13214779

**Published:** 2020-10-26

**Authors:** Yuzhe Wu, Yuntong Li, Conghui Yuan, Lizong Dai

**Affiliations:** Fujian Provincial Key Laboratory of Fire Retardant Materials, College of Materials, Xiamen University, Xiamen 361005, China; 20720150150066@stu.xmu.edu.cn (Y.W.); lyt@stu.xmu.edu.cn (Y.L.)

**Keywords:** coordination, metallosupramolecular polymer, active site, carbon materials, oxygen reduction reaction

## Abstract

Introduction of both nitrogen and transition metal elements into the carbon materials has demonstrated to be a promising strategy to construct highly active electrode materials for energy shortage. In this work, through the coordination reaction between Fe^3+^ and 1,3,5–tris(4–aminophenyl)benzene, metallosupramolecular polymer precursors are designed for the preparation of carbon flakes co-doped with both Fe and N elements. The as-prepared carbon flakes display wrinkled edges and comprise Fe_3_C nanoparticle and active site of Fe–N_x_. These carbon materials exhibit excellent electrocatalytic performance. Towards oxygen reduction reaction (ORR), the optimized sample has E_onset_ and E_half-wave_ of 0.93 V and 0.83 V in alkaline system, respectively, which are very close to that of Pt/C. This approach may offer a new way to high performance and low-cost electrochemical catalysts.

## 1. Introduction

In recent years, much attention has been focused on the development of facile and applicable methods to fabricate high-activity, low-cost oxygen reduction reaction (ORR) catalysts. This is significant to overcome the challenges in the commercialization and industrialization of hydrogen fuel cells. Nevertheless, Pt/C still acts as a main role in the commercialized ORR catalysts, even though it is relatively expensive [1,2,3,4]. Therefore, non-precious metal-based materials with high catalytic performance have attracted great research interest [5,6,7,8].

Indeed, transition metal elements have been widely introduced into the carbon materials to achieve high electrochemical performances [9,10,11,12]. Incorporation of transition metal elements into the carbon matrix generally relies on the design of composite precursors. For example, by using the reaction between salts (like iron(III) nitrate nonahydrate, nickel(II) nitrate hexahydrate, manganese(II) acetate tetrahydrate, and cobalt(II) nitrate hexahydrate) and graphite oxide, graphite oxide-metal-based precursors can be generated. Subsequently, a thermal procedure leads to the formation of Mn, Fe, Co, and Ni-coped graphene, which exhibited improved ORR performance [13]. Design of metal-organic precursors has been demonstrated to be a controllable method to generate transition metal elements doped carbon materials. It has been reported that precursors derived from the coordination between transition metals salts (MCl_x_, M = Cu, Ni, Co, Fe and Mn) and melamine/aniline, can be used to prepare transition metal doped carbon materials with enhanced ORR properties [14]. More interesting, utilization of metal-organic frameworks as precursors for the fabrication of transition metal-doped carbon materials has attracted great attention, because of the designable composition, pore structure, and tunable metal species [15]. Notably, co-doping of both iron and N elements in the carbon materials is of great advantage for improving the electrochemical properties, due to the generation of iron-nitrogen-carbon (Fe–N–C) active sites (Fe–N_x_ and Fe–C_x_) [16,17,18]. Usually, Fe–N–C catalysts were prepared by thermal cracking of iron macrocyclic polymers, iron-organic salts, or N-containing compounds [19,20]. The complicated synthetic process and high cost greatly limited their practical application [21,22]. So, it is still desirable to explore simple method to prepare Fe–N–C catalysts from commercial resources.

In this report, we show that a simple ligand-Fe^3+^ coordination strategy using commercial resources as starting materials, can create metallosupramolecular polymer precursors for the fabrication of Fe and N elements co-doped carbon materials with high ORR activity. The organic ligand adopted in this research is 1,3,5–tris(4–aminophenyl)benzene (denoted as TA) possessing three amine groups and conjugated structure. Because of the rigid structure of TA, the as-formed TA–Fe coordinative networks display uniform layer structure. After carbonization, carbon flakes with wrinkled edges (denoted as CNSs) can be easily generated. We have found that Fe_3_C nanoparticle and active site of Fe–N_x_ are formed in the carbon matrix, which can promote the ORR activity of the carbon materials. In addition, the feature of this research lies in the simple synthesis of CNSs. It can introduce Fe_3_C nanoparticle and Fe–N_x_ into the carbon matrix by one step. This synthetic route has certain universality and representativeness. It is also found that the appropriate molar ratio between amino ligand and Fe^3+^ was the most important factor that determines the activity of CNSs.

## 2. Experimental Section

### 2.1. Materials

1,3,5–Tris(4–aminophenyl)benzene, iron(III) chloride hexahydrate and dichloromethane were supplied by Aladdin Company (Shanghai, China) and directly used. Anhydrous ethanol, anhydrous methanol, KOH, and hydrochloric acid were purchased from Shanghai Chemical Reagent Industry (Shanghai, China). Nafion (5 wt%) was supplied by Sigma-Aldrich (Shanghai, China).

### 2.2. Catalysts Preparation

TA (0.3 g, 0.85 mmol) was firstly dissolved in 100 mL of dichloromethane with vigorous stirring for 3 h. Then, 1.38, 1.84, and 2.30 mL methanol solutions of Iron(III) chloride hexahydrate (100 mg/mL, 0.37 mmol/mL) were added into the solution drop by drop under vigorous stirring under N_2_ atmosphere. Thus, the corresponding molar ratios between TA and Fe^3+^ were 1.0:0.6, 1.0:0.8, and 1.0:1.0, respectively. The mixtures became yellow, and suspensions of metallosupramolecular polymers were formed after 12 h of reaction. The yellow powders were collected by centrifugation and washing with a mixed solvent of 30 mL dichloromethane and 30 mL anhydrous methanol three times. After drying in vacuum at 50 °C overnight, TA–Fe precursors were obtained (denoted as TA–0.6Fe, TA–0.8Fe and TA–1.0Fe). These precursors were then carbonized at 850 °C for 2 h in argon gas with a heating rate of 5 °C/min to generate the carbon materials. The carbon materials were washed by 6 M HCl for 5 h and 100 mL of ultrapure water three times at room temperature, then dried by freeze-drying overnight. A second carbonization was performed at 850 °C for 2 h in argon gas with a heating rate of 10 °C/min to get the target carbon materials (denoted as CNS–0.6Fe, CNS–0.8Fe and CNS–1.0Fe).

### 2.3. Characterization

The scanning electron microscopy (SEM) images were observed through an SU-70 microscopy instrument (HITACHI, Tokyo, Japan). The FTIR measurements were tested by an AVATAR 360 FTIR (Nicolet Instrument, Tokyo, Japan) at room temperature. The powder X-ray diffraction (XRD) patterns were measured through a Desktop X-ray Diffractometer ((Rigaku, Tokyo, Japan) using Cu (600 W) Kα radiation. Raman spectra were tested by a Labram HR800 Evolution (Horiba, Lille, France). The X-ray photoelectron spectroscopy (XPS) were tested by PHI Quantum-2000 photoelectron spectrometer (Physical Electronics, Inc., Chanhassen, MN, USA). The pore volume and Brunauer–Emmett–Teller (BET) were taken through an ASAP 2460 system (Norcross, GA, USA). Electron paramagnetic resonance (EPR) experiments were conducted on an electron spin resonance spectrometer (Bruker EMX-10/12, Bruker UK, Coventry, UK) at 90 K. Transmission electron microscopic (TEM) measurements were performed using a JEM-2100 microscope (JEOL, Tokyo, Japan). The elemental energy-dispersive X-ray spectroscopy (EDX) were obtained by using a FEI TECNAI F20 microscope (Hillsboro, OR, USA).

### 2.4. Electrochemical Measurements

The electrochemical experiments were conducted on an electrochemical workstation (CHI 760E, Chenhua, Shanghai, China), by using the typical three-electrode system. A standard rotating disk electrode with a glassy carbon disk (5 mm in diameter) was applied as working electrode. Before test, CNS–0.6Fe, CNS–0.8Fe, and CNS–1.0Fe (5.0 mg) were dispersed in 1.0 mL of homogeneous solvent with 500 μL of anhydrous ethanol, 450 μL of H_2_O and 50 μL of 5 wt% Nafion. The above newly made slurry (4.5 μL) was carefully dropped onto a glassy carbon electrode as working electrode. The ORR performance was tested in newly made KOH aqueous solution (0.1 mol/L) at room temperature. Pt foil and Ag/AgCl (KCl saturation) electrode were separately applied as counter electrode and reference electrode. The potential in this study was relative to the Ag/AgCl electrode.

## 3. Results and Discussion

### 3.1. Characterization of Metallosupramolecular Polymer Precursors

The synthetic process of precursor is very simple. As shown in Scheme 1, metallosupramolecular polymer precursors are generated from the direct reaction between commercially available resources. In a dichloromethane solution, the high coordination affinity between TA and Fe^3+^ can easily induce the formation precipitates. After a pyrolysis treatment of the precursors, carbon materials comprising both Fe_3_C nanoparticle and Fe–N_x_ active site can be fabricated easily.

The typical SEM image of TA–0.8Fe is displayed in Figure 1a, from which sheet-like morphology can be observed. The coordination reaction between TA and Fe^3+^ was verified by FTIR as shown in Figure 1b. The peaks located at 3350–3343 cm^−1^ correspond to the characteristic signals of amino groups of TA, and the peaks at 845, 685, and 600 cm^−1^ are derived from iron(III) chloride hexahydrate. Comparing the spectra of precursors with TA, the characteristic peak of amino groups shifts and becomes broaden. Also, the characteristic peaks of Fe^3+^ in the precursors were evidently weakened. These results indicate the coordination between Fe^3+^ and TA [23].

In the EPR spectra (Figure 2a), all samples show a small radical signal of amino group at *g’* = 2.00. For TA–0.6Fe, TA–0.8Fe, and TA–1.0Fe, the representative signal of Fe^3+^ at *g’* = 4.25 can be observed, indicating the presence of Fe^3+^ in the precursor [24]. The XPS survey spectra of TA, TA–0.6Fe, TA–0.8Fe, and TA–1.0Fe are displayed in Figure 2b. The representative signals of Fe 2p locate at 714.4 ± 0.1 and 725.4 ± 0.1 eV, which can be attributed to the binding energies of 2p_3/2_ and 2p_1/2_ orbitals of Fe^3+^, respectively. Figure 2c–f shows the high-resolution XPS spectra of N 1s of TA and the precursors. A signal at 399.4 ± 0.1 eV is attributed to the amino group. In the case of TA–0.6Fe, TA–0.8Fe, and TA–1.0Fe, a peak at 401.6 ± 0.1 eV is attributed to amino group perturbed by Fe^3+^ [25], which is helpful for the formation of Fe–N–C active site during carbonization. As showed in Appendix A, TA–0.8Fe has higher content of Fe–NH_2_ than TA–0.6Fe and TA–1.0Fe, which may result in more content of Fe–N_x_ active site after carbonization.

### 3.2. Structure and Composition of CNSs

After twice carbonization at 850 °C, the as obtained CNSs can maintain the lamellar structure, but the edges become wrinkled (Figure 3a,b and Appendix A). This structure may be helpful for the direct contact between the active sites with the oxygen, thus improving the electrocatalytic activity of carbon materials. Notably, CNS–0.8Fe possesses the most uniform lamellar morphology (Figure 3b). The high-resolution TEM images of CNS–0.8Fe show clear inter-planar distance of 0.201 nm derived from the (031) plan of Fe_3_C nanoparticle (Figure 3c,d). The outer carbon coating on the Fe_3_C nanoparticle has a good lattice structure with a spacing of 0.34 nm, corresponding to the (002) plan of graphitic carbon (Figure 3c,d). The outer layer of graphitized carbon on Fe_3_C nanoparticles has good electrical conductivity. During the ORR catalytic process, Fe_3_C nanoparticles may not contact with electrolyte directly, but can play the catalytic role indirectly through the outer layer of graphitized carbon to improve the catalytic activity. The Fe_3_C nanoparticle generated in CNSs can improve the ORR activity of carbon materials, which was confirmed already [26]. Figure 3e–i gives the dark-field TEM image and EDX mapping of CNS–0.8Fe. Obviously, elements of C, N, Fe, and O are homogeneously dispersed all through the carbon materials.

Figure 4a illustrates the Raman spectra of the CNSs. For CNS–0.6Fe, CNS–0.8Fe and CNS–1.0Fe, the intensity ratios between D band (1340 cm^−1^) derived from disordered graphitic structure and G band (1571 cm^−1^) derived from ordered carbon structure are calculated to be 0.97, 0.95, and 1.04, respectively. This result indicates that the graphitic degree of CNS–0.8Fe is higher than that of the other samples. The crystalline structures of CNSs were evaluated by the XRD. As displayed in Figure 4b, a broad diffraction peak at about 25° is attributed to the (002) plane of ordered graphitic structure. Moreover, the XRD results clearly confirm that the iron element is remained in the carbon matrix as a Fe_3_C form. All the diffraction peaks are in good agreement with that of the Fe_3_C (JCPDS Card No.65−2413). These results, in combination with the high-resolution TEM images, prove the presence of Fe_3_C nanoparticle in the CNSs catalysts, which may promote the ORR activity of carbon materials [27].

The pore character of CNSs was characterized through the physisorption of nitrogen at 77 K. As shown in Figure 5, all samples show well-developed micro-pore and mesoporous-pore structures. Table 1 shows the corresponding information about BET surface area as well as total pore volumes of CNSs. The surface areas of CNS–0.6Fe, CNS–0.8Fe, and CNS–1.0Fe are 167.86, 196.20, and 145.20 m^2^·g^−1^, with relevant pore volumes of 0.16, 0.17, and 0.17 cm^3^·g^−1^, respectively. Obviously, surface areas of CNSs are resulted from both micro-pore and mesoporous-pore structures. The CNS–0.8Fe has a higher BET surface area than CNS–0.6Fe and CNS–1.0Fe. We consider that the large surface area of CNS–0.8Fe is attributed to the moderate crosslinking degree of the metallosupramolecular polymer networks.

The XPS survey spectra of the CNSs are shown in Figure 6a. Also, the high-resolution XPS spectra of C 1s, N 1s, and Fe 2p of CNS–0.8Fe are displayed Figure 6b–d. The C 1s signal is split into three representative peaks at 284.4 ± 0.1 (C=C, C–C), 285.4 ± 0.1 (C–O, C–N), and 288.2 ± 0.1 eV (C=O). Four peaks of N 1s signal at 398.5 ± 0.1, 399.5 ± 0.1, 401.2 ± 0.1, and 404.5 ± 0.1 eV are respectively belong to the pyridinic N, Fe–N_x_, graphitic N, and oxidized N. Notably, Fe–N_x_ and pyridinic N are recognized to be promising for the improvement of ORR activity [28,29]. The N 1s spectra of CNS–0.6Fe and CNS–1.0Fe are also showed in Appendix A. For the Fe 2p spectrum, the peak located at 725.4 ± 0.1 eV is assigned to the binding energy of Fe^3+^ for the 2p_1/2_ band, and the peak of Fe^2+^ is detected at 723.2 ± 0.1 eV for the 2p_1/2_ band. Another two peaks at 714.4 ± 0.1 and 710.5 ± 0.1 eV can be respectively attributed to the binding energies of 2p_3/2_ orbitals of Fe^3+^ as well as Fe^2+^ species. The last signal at 719.6 ± 0.1 eV is the satellite peak. These XPS results, in combination with the XRD results, clearly confirm that the presence of Fe_3_C nanoparticle, and active sites of Fe–N_x_ and pyridinic N in the carbon matrix of CNSs. Moreover, as listed in Table 2, the pyridinic N and Fe–N_x_ contents of CNS–0.8Fe are 0.28 and 0.43 at.%, respectively, which are much higher than that of CNS–0.6Fe (0.17 and 0.20 at.%) and CNS–1.0Fe (0.25 and 0.22 at.%). This result indicates that CNS–0.8Fe may have more active sites towards ORR. In summary, the Fe and N elements co-doping effect leads to the generation of both Fe–N_x_ active site and Fe_3_C nanoparticle when carbonization, which can greatly improve the catalytic activity towards ORR. That is the main mechanism and contribution of the co-doping effect of Fe and N elements in CNSs.

### 3.3. ORR Performance of CNSs

The CV curves of CNSs were tested in both Ar or O_2_-saturated 0.1 M KOH solution (Figure 7a). The samples show no reduction peak in Ar-saturated solution but show a typical reduction peak when changed into O_2_-saturated solution. The double-layer capacitance of the three samples was researched, which are showed in Appendix A. The CV area of CNS–0.8Fe in Ar is larger than CNS–0.6Fe and CNS–1.0Fe. This indicates that CNS–0.8Fe owns larger electrochemically active surface area than the other two samples, which is helpful for improving the catalytic activity of ORR. Figure 7b list the LSV curves of CNSs. The onset potential (E_onset_) of CNS–0.6Fe, CNS–0.8Fe, and CNS–1.0Fe for ORR are 0.89, 0.93, and 0.88 V vs. RHE, separately. The half-wave (E_half-wave_) of the CNS–0.6Fe, CNS–0.8Fe and CNS–1.0Fe are 0.78, 0.83, and 0.80 V vs. RHE, separately. The above results were confirmed by LSV tests in the newly made O_2_-saturated solution at the scanned rate of 10 mV/s with the fixed rotation speed of 1600 rpm. As a control experiment, the ORR activity of Pt/C catalyst was also tested with the same experimental condition (Figure 7b). Apparently, the E_onset_ (0.93 V) and E_half-wave_ (0.83 V) values of CNS–0.8Fe are very close to Pt/C catalyst (E_onset_ = 0.95 V and E_half-wave_ = 0.85 V). The Tafel plots of CNSs were tested (Appendix A). The Tafel slopes of CNS–0.6Fe and CNS–1.0Fe are 89 and 84 mV·dec^−1^, respectively. However, a Tafel slope of 73 mV·dec^−1^ is detected for CNS–0.8Fe, which is lower than that of Pt/C (78 mV·dec^−1^) and directly indicating that CNS–0.8Fe owns faster ORR kinetics.

The evidently improved electrocatalytic performance of CNS–0.8Fe among the three samples can be explained by the following four reasons. First, CNS–0.8Fe has relative higher specific surface area in comparison with CNS–0.6Fe and CNS–1.0Fe, thus resulting in the expose of more active sites. Second, CNS–0.8Fe possesses a better developed lamellar structure than CNS–0.6Fe and CNS–1.0Fe as indicated by the TEM images, which is beneficial for the contact between active sites and oxygen molecules during ORR process. Third, the calculated I_D_/I_G_ value testified that the graphitic degree of CNS–0.8Fe is higher than the others, thus endowing this sample with a better electrical conductivity. Fourth and most importantly, CNS–0.8Fe possesses higher pyridinic N and Fe–N_x_ contents than CNS–0.6Fe and CNS–1.0Fe, which can provide more active sites towards ORR, thus greatly enhancing the catalytic activity.

The LSV curves of CNSs were also collected with different rotation rates. The current density of CNS–0.6Fe, CNS–0.8Fe, and CNS–1.0Fe increase gradually when consecutively changing the rotation speeds from 400 to 1600 rpm, as listed in Appendix A. Probably, the shortening of the diffusion distance directly leads to this regular phenomenon. To further explore the reaction kinetics of the ORR process, rotating ring disk electrode (RRDE) experiments were performed to calculate the generation of HO_2_^−^ also with electron transfer numbers (n) values. CNS–0.6Fe, CNS–0.8Fe, CNS–1.0Fe, and the commercial Pt/C displayed higher disk current but minor ring current, as shown in Figure 8a. With the increase of potential from 0.2 to 0.5 V, the corresponding HO_2_^−^ yield ranges of CNS–0.6Fe, CNS–0.8Fe, CNS–1.0Fe, and Pt/C catalyst are 10.57 to 11.90%, 3.99 to 6.22%, 4.66 to 5.93%, and 1.81 to 2.71%, as shown in Figure 8b. Also, the corresponding n values of CNS–0.6Fe, CNS–0.8Fe, CNS–1.0Fe, and Pt/C catalyst are 3.67 to 3.78, 3.87 to 3.92, 3.88 to 3.90, and 3.94 to 3.96 as shown in Figure 8c. So, these results just could indicate that CNSs catalyze ORR by the typical dominant four-electron transfer pathway [30,31,32,33].

Taking CNS–0.8Fe as an example, the durability of CNSs was evaluated at 1600 rpm with consecutive 1000 cycles of CV scan in O_2_-saturated 0.1 M KOH solution (Appendix A). The decrease of the onset and half-wave potentials is not evident. The LSV curves recorded before and after 1000 cycles reveal negative shifts of E_half-wave_ of 7 mV for CNS–0.8Fe, which is lower than that of Pt/C (12 mV) as reported [21]. This result indicates that CNS–0.8Fe is relatively stable for ORR. The relevant crossover effects tests were conducted by taking CNS–0.8Fe as an example through chronoamperometric measurement to evaluate the catalytic selectivity of the catalysts. The methanol oxidation reaction resulted that the current density of Pt/C catalyst directly decreased at once when scrupulously adding 3.0 M methanol, as shown in Figure 9. However, this was not the case for CNS–0.8Fe, as the current density of that did not show evident change (Figure 9). These results confirmed that CNSs have good catalytic selectivity for ORR [34,35,36].

## 4. Conclusions

In summary, we prepared a new type of Fe and N co-doped carbon materials through a simple and effective method in one step. Direct coordination between amino ligand and Fe^3+^ could easily afford metallosupramolecular polymer precursors. After two carbonization processes, carbon flakes with wrinkled edges and active site of Fe–N_x_ and Fe_3_C nanoparticle were fabricated. The catalytic activity of the carbon materials towards ORR were detailed investigated. The carbon material of CNS–0.8Fe possessed E_onset_ = 0.93 V and E_half-wave_ = 0.83 V vs. RHE in alkaline system, which were comparable to Pt/C catalyst. The ligand TA and Fe^3+^ could generate more content of Fe–NH_2_ in the precursor at a proper proportion through the coordination reaction and further led to the generation of more content of Fe–N_x_ active site when carbonization. So, the appropriate molar ratio between amino ligand and Fe^3+^ was the most important factor that determined the activity of CNSs. We considered that this simple method and conclusion might be of practical interest for the exploration of electrocatalysts with excellent ORR activity.

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
