# Peer review of "A Facile Method for the Generation of Fe3C Nanoparticle and Fe-Nx Active Site in Carbon Matrix to Achieve Good Oxygen Reduction Reaction Electrochemical Performances"

_materials, 2020, doi:10.3390/ma13214779_

Round 1
Reviewer 1 Report
In this draft, the authors describe a Fe-N-C catalyst containing iron carbide and FeNx sites for ORR in alkali media. However, several key points are missing to fully describe the catalyst they have reported:
- Both iron carbide and FeNx are present in the Fe-N-C catalyst. Which is the main active site toward ORR? Is there any synergy between the two kinds of active sites? The authors mentioned that the carbide will enhance the ORR activity; while no evidence or discussion is provided to support this. Particularly, the amount of FeNx sites reaches a peak when increasing the iron loading in the precursor (TA-0.8Fe). Further increasing the iron loading in the precursor will decrease the density of FeNx. Is this means the extra Fe has transformed into carbide or other iron species? When describing the activity trend, only the amount of FeNx and pyridinic N are considered, but not iron carbides.
- According to the TEM images, the carbide species is covered by a thick carbon layer (>5nm, Figure 3d). So how can the carbides be active toward ORR? More discussion explanation is required.
- The Raman suggests that CNS-0.8Fe is more graphitic than the other two samples, while the BET suggests it is more porous. However, generally more graphitic carbon is less porous. The authors should explain why their sample has the opposite trend. Moreover, According to Figure 5a, the differences in the BET surface area of the three samples should be more prominent than those reported in Table 1. Please double-check the y-axis of Figure 5a.
- Also please check the y-axis of Figure 7a. The current density of a CV in Ar should not be far above or below 0. Also, the authors should compare the double-layer capacitance of the three samples. This should be more relevant since it represents the electrochemically active surface area.
Author Response
Answers to Comments of Reviewer 1: In this draft, the authors describe a Fe-N-C catalyst containing iron carbide and Fe-Nx sites for ORR in alkali media. However, several key points are missing to fully describe the catalyst they have reported:
- Both iron carbide and Fe-Nx are present in the Fe-N-C catalyst. Which is the main active site toward ORR? Is there any synergy between the two kinds of active sites? The authors mentioned that the carbide will enhance the ORR activity; while no evidence or discussion is provided to support this. Particularly, the amount of Fe-Nx sites reaches a peak when increasing the iron loading in the precursor (TA-0.8Fe). Further increasing the iron loading in the precursor will decrease the density of Fe-Nx. Is this means the extra Fe has transformed into carbide or other iron species? When describing the activity trend, only the amount of Fe-Nx and pyridinic N are considered, but not iron carbides.
Answer: We thank the reviewer #1 very much for this comment. Fe-Nx is the main active site toward ORR for CNSs. During the carbonization of procursors, iron is reduced to metal Fe by thermal treatment. The diffusion of free N atom can contact with highly chemically active Fe, thus forming Fe-Nx firstly. With the increase of temperature, the excessive Fe will react with C atom to form the Fe3C nanoparticle. Therefore, the formation of Fe3C nanoparticle is accompanied with the formation of Fe-Nx. Fe3C nanoparticles can enhance the oxygen reduction activity of Fe-Nx which has been reported before (J. Am. Chem. Soc. 2016, 138(10): 3570-3578). With the increasing the iron loading in the precursors, too much iron affects the coordination of iron with ligand TA and lead to an decreased content of Fe-Nx. The extra iron was removed by 6 M HCl for 5 h after the first carbonization. In terms of current technology, the content of Fe3C nanoparticles cannot be calculated, so it is unable to evaluate the direct impact about the content of Fe3C nanoparticle to the ORR activity for CNSs. The advantage of our method is that it can generate Fe-Nx and Fe3C nanoparticle at the same time by one step coordination reaction through amino ligand TA and Fe3+, and introduce Fe-Nx and Fe3C nanoparticle doped into the CNSs, which is beneficial to the improvement of ORR activity. Hope reviewer #1 will approve of our answer.
- According to the TEM images, the carbide species is covered by a thick carbon layer (>5 nm, Figure 3d). So how can the carbides be active toward ORR? More discussion explanation is required.
Answer: We thank the reviewer #1 very much for this comment. Fe3C nanoparticle can promote the ORR activity in some extent, which has been reported by many literatures (Journal of Materials Chemistry A, 2014, 3(4): 1752-1760; Journal of Power Sources, 2018, 78: 491-498; Carbon, 2019, 150: 93-100; J. Am. Chem. Soc. 2016, 138(10): 3570-3578.). On page 5, line 153-157 in the revised manuscript, “The outer layer of graphitized carbon on Fe3C nanoparticles has good electrical conductivity. During the ORR catalytic process, Fe3C nanoparticles may not contact with electrolyte directly, but can play the catalytic role indirectly through the outer layer of graphitized carbon to improve the catalytic activity. The Fe3C nanoparticle generated in CNSs can improve the ORR activity of carbon materials, which was confirmed already [26].” was added. That is the mechanism of action about the Fe3C nanoparticle. Detailed information can be seen on page 5, line 153-157 in the revised manuscript. Hope reviewer #1 will approve of our answer.
- The Raman suggests that CNS-0.8Fe is more graphitic than the other two samples, while the BET suggests it is more porous. However, generally more graphitic carbon is less porous. The authors should explain why their sample has the opposite trend. Moreover, According to Figure 5a, the differences in the BET surface area of the three samples should be more prominent than those reported in Table 1. Please double-check the y-axis of Figure 5a.
Answer: We thank the reviewer #1 very much for this comment. It is very sorry for that we made a mistake. The y-axis of Figure 5a was mislabeled when drawing the Figure 5. Now we have revised the y-axis of Figure 5a, the new result is right. Detailed information can be seen in Figure 5a on page 6, line 180-182 in the revised manuscript. Moreover, i) the specific surface areas of CNS-0.6Fe and CNS-1.0Fe are close to CNS-0.8Fe, the difference is not too great. ii) The larger surface area of CNS-0.8Fe is attributed to the moderate crosslinking degree of the metallosupramolecular polymer networks. Less iron is not good for the coordination of iron with ligand TA, but more iron will affect the coordination of iron with ligand TA. These two reasons will lead to the decrease of surface area of CNS-0.6Fe and CNS-1.0Fe. So, an appropriate coordination proportion of TA-0.8Fe leads to the highest crosslinking degree of the metallosupramolecular polymer networks in precursors and then results into an increased surface area of CNS-0.8Fe. On the other hand, the proper iron content and the appropriate coordination ratio between amino ligand TA and Fe3+ maybe the main factor to promote the graphitization of CNS-0.8Fe. These factors maybe are the main reasons that led CNS-0.8Fe owned larger larger surface area and higher graphitization. Hope reviewer #1 will approve of our answer.
- Also please check the y-axis of Figure 7a. The current density of a CV in Ar should not be far above or below 0. Also, the authors should compare the double-layer capacitance of the three samples. This should be more relevant since it represents the electrochemically active surface area.
Answer: We thank the reviewer #1 very much for this comment. It is very sorry for that we made a mistake. The y-axis of Figure 7a was mislabeled when drawing the Figure 7. Now we have revised the original Figure 7a on page 8, line 244-246, the new result is right. And we have added the double-layer capacitance study in the revised manuscript. In the supplementary information, on page 3, the Figure S3 of double-layer capacitance was added.
On page 8, line 230-233, “The double-layer capacitance of the three samples was researched, which are showed in Figure S3. The CV area of CNS-0.8Fe in Ar is larger than CNS-0.6Fe and CNS-1.0Fe. This indicates that CNS-0.8Fe owns larger electrochemically active surface area than the other two samples, which is helpful for improving the catalytic activity of ORR.” was added in the revised manuscript. Detailed information can be seen in the supplementary information of Figure S3 and page 8, line 230-233 in the revised manuscript. Hope reviewer #1 will approve of our answer.

Reviewer 2 Report
Please find attacted the revisions

Author Response
Answers to Comments of Reviewer 2:
- Introduction section needs enhancement in order the novelty and the contribution
of the current work to be clarified.
Answer: We thank the reviewer #2 very much for this suggestion. We have revised the introduction section carefully. On page 2, line 60-64, “In addition, the feature of this research lies in the simple synthesis of CNSs. It can introduce Fe3C nanoparticle and Fe-Nx into the carbon matrix by one step. This synthetic route has certain universality and representativeness. It is also found that the appropriate molar ratio between amino ligand and Fe3+ was the most important factor that determines the activity of CNSs.” was added. And detailed information can be seen on page 2, line 60-64 in the revised manuscript. Hope reviewer #2 will approve of our answer.
- Each section should start with text and not with Figure. For example, 3.2 sub-section needs revision.
Answer: We thank the reviewer #2 very much for this suggestion. We have adjusted the format of the manuscript according to the requirements of the journal, so it can better meet the conditions for publication. Detailed information can be in the revised manuscript. Hope reviewer #2 will approve of our answer.
- In comparison to the conventional catalysts (such as Pt/C, Pd/C) results for the ORR, how the authors evaluate the performance of the as-investigated electrocatalysts?
Answer: We thank the reviewer #2 very much for this comment. CNS-0.8Fe belongs to non-precious metal catalyst, has the advantage of low cost, and the catalytic activity of CNS-0.8Fe is very close to that of commercial Pt/C, but there is still a small gap between them. The catalytic selectivity of CNS-0.8Fe towards ORR is better than commercial Pt/C. The synthesis route of CNSs is also very simple and efficient too. In conclusion, our research can provide a new reference and thought for the research of new type catalysts in the future work. Hope reviewer #2 will approve of our answer.
- The authors should compare their results with relative works in literature (for example. i) Single iron atoms coordinated to g-C3N4on hierarchical porous N-doped carbon polyhedra as a high-performance electrocatalyst for the oxygen reduction reaction, ii) Synthesis of nitrogen-doped mesoporous carbon nanosheets for oxygen reduction electrocatalytic activity enhancement in acid and alkaline media, etc)
Answer: We thank the reviewer #2 very much for this suggestion. We have carefully checked the literatures suggested by the reviewer #3. These literatures are significant, and they detailed studied the doping effect of heteroatoms towards ORR. Our work focuses on the co-doping of both Fe and N into carbon materials, it has some differences from the previous study. Compared to these literatures, our advantage is that the property of CNS-0.8Fe is close to them, but our synthetic route is very simple and efficient, which provides scalable strategy for the preparation of Fe and N co-doped carbon catalysts. Hope reviewer #2 will approve of our answer.
- A detailed kinetic analysis (Tafel plots) is necessary to be included into the manuscript.
Answer: We thank the reviewer #2 very much for this comment. We have added the Tafel plots study in the revised manuscript. In the supplementary information of page 3, the Figure S4 of Tafel plots was added. On page 8, line 240-243 in the revised manuscript, “The Tafel plots of CNSs were tested (Figure S4). The Tafel slopes of CNS-0.6Fe, and CNS-1.0Fe are 89 and 84 mV dec-1, respectively. But a Tafel slope of 73 mV dec-1 is detected for CNS-0.8Fe, which is lower than that of Pt/C (78 mV dec-1) and directly indicating that CNS-0.8Fe owns faster ORR kinetics.” was added. Detailed information can be seen in the supplementary information of Figure S4 and page 8, line 240-243 in the revised manuscript. Hope reviewer #2 will approve of our answer.
- The possible mechanism and the contribution of the Fe and N into the carbon to the ORR, should be also further discussed.
Answer: We thank the reviewer #2 very much for this comment. The metallosupramolecular polymer precursors are formed by the coordination between amino group and Fe3+. During the carbonization of procursors, iron is reduced to metal Fe by thermal treatment. The diffusion of free N atom can contact with highly chemically active Fe, thus forming Fe-Nx firstly. With the increase of temperature, the excessive Fe will react with C atom to form the Fe3C nanoparticle. Therefore, the formation of Fe3C nanoparticle is accompanied with the formation of Fe-Nx. The advantage of our method is that it can generate Fe-Nx and Fe3C nanoparticle at the same time by one step coordination reaction through amino ligand TA and Fe3+, and introduce Fe-Nx and Fe3C nanoparticle doped into the CNSs, which is beneficial to the improvement of ORR activity. And these are the mechanism and contribution of the Fe and N into the CNSs to the ORR.
On page 8, line 222-225 in the revised manuscript, “In summary, the Fe and N elements co-doping effect leads to the generation of both Fe-Nx active site and Fe3C nanoparticle when carbonization, which can greatly improve the catalytic activity towards ORR. That is the main mechanism and contribution of the co-doping effect of Fe and N elements in CNSs.” was added. Detailed information can be seen on page 8, line 222-225 in the revised manuscript. Hope reviewer #2 will approve of our answer.
- Moreover durability and stability over time tests are necessary to be conducted.
Answer: We thank the reviewer #2 very much for this suggestion. We have added the durability test experiment of CNS-0.8Fe in the revised manuscript. In the supplementary information of page 4, the Figure S6 of durability test was added.
On page 9, line 279-283 in the revised manuscript, “Taking CNS-0.8Fe as an example, the durability of CNSs was evaluated at 1600 rpm with consecutive 1000 cycles of CV scan in O2-saturated 0.1 M KOH solution (Figure S6). The decrease of the onset and half-wave potentials is not evident. The LSV curves recorded before and after 1000 cycles reveal negative shifts of Ehalf-wave of 7 mV for CNS-0.8Fe, which is lower than that of Pt/C (12 mV) as reported [21]. This result indicates that CNS-0.8Fe is relatively stable for ORR.” was added. Detailed information can be seen in the supplementary information of Figure S6 and page 9, line 279-283 in the revised manuscript. Hope reviewer #2 will approve of our answer.

Reviewer 3 Report
This is a very interesting study and also well written.
Just two comments:
- please use in the abstract "as-prepared".
- in conclusions, please explain what TA means. Maybe some readers will not understand immediatly.
Author Response
The page number and line number are according to the revised manuscript. New changes to the manuscript have been highlighted with red color.
Answers to Comments of Reviewer 3: This is a very interesting study and also well written. Just two comments:
- Please use in the abstract "as-prepared".
Answer: We thank the reviewer #3 very much for this suggestion. On page 1, line 16, in the revised manuscript, “as prepared” was changed into “as-prepared”. Detailed information can be seen on page 1, line 16 in the revised manuscript. Hope reviewer #3 will approve of our answer.
- In conclusions, please explain what TA means. Maybe some readers will not understand immediatly.
Answer: We thank the reviewer #3 very much for this suggestion. On page 10, line 293 and line 301 in the revised manuscript, “TA” was changed into “amino ligand”. So it can help readers have a better understanding of the meaning. Detailed information can be seen on page 10, line 293 and line 301 in the revised manuscript. Hope reviewer #3 will approve of our answer.

Reviewer 4 Report
Authors deal with the preparation and characterization of Fe-N-C characterized by Fe3C nanoparticles and Fe-Nx active sites. Phisico-chemical characterization is well organized and appropriate and electrochemical results in alkaline media are promising for the ORR of this non –precious metal electrocatalyst.
However, the article could be published in the journal if minor revisions are addressed:
- Why do the authors denominate the samples as CNS if S is not a precursor of the synthesis?
- In the scheme 1, TA is not visible properly. Please, improve the image.
- Methanol tolerance of these electrocatalysts is well known. The authors should add some references (for example Catalysts 2018, 8, 650; https://doi.org/10.3390/catal8120650; ACS Catal.2020, 10, 14, 7475–7485 https://doi.org/10.1021/acscatal.0c01288; Journal of Power Sources 250:279-285 DOI: 1016/j.jpowsour.2013.11.011)
- A durability study should be included to have greater visibility.

Author Response
Answers to Comments of Reviewer 4: Authors deal with the preparation and characterization of Fe-N-C characterized by Fe3C nanoparticles and Fe-Nx active sites. Phisico-chemical characterization is well organized and appropriate and electrochemical results in alkaline media are promising for the ORR of this non-precious metal electrocatalyst. However, the article could be published in the journal if minor revisions are addressed:
- Why do the authors denominate the samples as CNS if S is not a precursor of the synthesis?
Answer: We thank the reviewer #4 very much for this comment. S does not represent sulfur element, but means that of “carbon flakes”. On page 2, line 58, we named the samples in this way is because that CNS can better represent the meaning of this kind of materials. Hope reviewer #4 will approve of our answer.
- In the scheme 1, TA is not visible properly. Please, improve the image.
Answer: We thank the reviewer #4 very much for this suggestion. We have made some changes to the scheme 1 to make it clearer, specific revisions can be seen in the manuscript of the scheme 1 on page 3, line 115-116 in the revised manuscript. Hope reviewer #4 will approve of our answer.
- Methanol tolerance of these electrocatalysts is well known. The authors should add some references (for example Catalysts 2018, 8, 650; https://doi.org/10.3390/catal8120650; ACS Catal. 2020, 10, 14, 7475-7485 https://doi.org/10.1021/acscatal.0c01288; Journal of Power Sources 250:279-285 DOI: 10.1016/j.jpowsour.2013.11.011).
Answer: We thank the reviewer #4 very much for this suggestion. We have added these representative literatures as important references in the manuscript to make it more representative. Detailed information can be seen on page 10, line 289 and on page 12, line 411-419 in the revised manuscript. Hope reviewer #4 will approve of our answer.
- A durability study should be included to have greater visibility.
Answer: We thank the reviewer #4 very much for this suggestion. We have added the durability test experiment of CNS-0.8Fe in the revised manuscript. In the supplementary information of page 4, the Figure S6 of durability test was added.
On page 9, line 279-283 in the revised manuscript, “Taking CNS-0.8Fe as an example, the durability of CNSs was evaluated at 1600 rpm with consecutive 1000 cycles of CV scan in O2-saturated 0.1 M KOH solution (Figure S6). The decrease of the onset and half-wave potentials is not evident. The LSV curves recorded before and after 1000 cycles reveal negative shifts of Ehalf-wave of 7 mV for CNS-0.8Fe, which is lower than that of Pt/C (12 mV) as reported [21]. This result indicates that CNS-0.8Fe is relatively stable for ORR.” was added. Detailed information can be seen in the supplementary information of Figure S6 and page 9, line 279-283 in the revised manuscript. Hope reviewer #4 will approve of our answer.

Round 2
Reviewer 1 Report
The authors have answered the questions properly. I would recommend its publication now.
Reviewer 2 Report
In the current form the manuscript is appropriate to be published.